# A Mini Review on Molecules Inducing Caspase-Independent Cell Death: A New Route to Cancer Therapy

**DOI:** 10.3390/molecules27196401

**Published:** 2022-09-28

**Authors:** Kakali Bhadra

**Affiliations:** Department of Zoology, University of Kalyani, Nadia, Kalyani 741235, India; kbzooku17@klyuniv.ac.in or kakali_bhadra2004@yahoo.com; Tel.: +033-2582-8750 (ext. 8220)

**Keywords:** caspase-dependent programmed cell death, caspase-independent programmed cell death, apoptosis, apoptosis inducing factor AIF

## Abstract

Most anticancer treatments trigger tumor cell death through apoptosis, where initiation of proteolytic action of caspase protein is a basic need. But under certain circumstances, apoptosis is prevented by the apoptosis inhibitor proteins, survivin and Hsp70. Several drugs focusing on classical programmed death of the cell have been reported to have low anti-tumorogenic potency due to mutations in proteins involved in the caspase-dependent programmed cell death with intrinsic and extrinsic pathways. This review concentrates on the role of anti-cancer drug molecules targeting alternative pathways of cancer cell death for treatment, by providing a molecular basis for the new strategies of novel anti-cancer treatment. Under these conditions, active agents targeting alternative cell death pathways can be considered as potent chemotherapeutic drugs. Many natural compounds and other small molecules, such as inorganic and synthetic compounds, including several repurposing drugs, are reported to cause caspase-independent cell death in the system. However, few molecules indicated both caspase-dependent as well caspase-free cell death in specific cancer lines. Cancer cells have alternative methods of caspase-independent programmed cell death which are equally promising for being targeted by small molecules. These small molecules may be useful leads for rational therapeutic drug design, and can be of potential interest for future cancer-preventive strategies.

## 1. Introduction

For many years, programmed cell death (PCD), which generally involves proteases like caspases, has been used as a focal killers of cells, and has played a significant role in apoptosis [1,2,3]. Caspases are aspartate-directed cysteine-dependent proteases that exist as multi-member families in metazoans such as humans and Drosophila [4,5,6]. However, *Drosophila* initiator caspase (Dronc), an ortholog of human caspase, functions in both apoptotic and non-apoptotic pathways by controlling the mechanism of ubiquitylation of the ortholog protein [7]. Caspases remain as inactive zymogens in cells and undergo a succession of catalytic processes that lead to the process of cell death. Active caspase during apoptosis causes proteolytic degradation of cellular structures, such as the cell junctions, cytoskeleton, Golgi apparatus, mitochondria, endoplasmic reticulum and the nucleus, ultimately causing programmed cell death [8]. However, an increasing number of recent studies support the existence of alternative cell death pathways. Although there are some differences of opinion regarding the terminology of the various forms of PCD, in the majority of the current literature, the term apoptosis is solely used for caspase-dependent death of the cell [2,3,9,10,11,12,13,14,15,16]. With the emerging evidence in the literature [1,3,17,18,19,20,21], cell death has been divided and modified into different forms. According to category, these are: (1) Classical apoptosis or caspase-dependent programmed cell death (extrinsic and intrinsic apoptosis) (Figure 1) Classical, caspase-dependent apoptosis through the extrinsic pathway [22,23] is initiated by receptor binding and the recruitment of FADD- and TRADD-like adapter molecules to the death domain (DD). This is followed by the binding of procaspase-8 to the death effector domains (DED) of these adaptor molecules, then activation of caspase-8 takes place. Caspase-8 then either directly activates caspases-3 and -7 or activates the cytosolic protein, Bid (Bcl2 protein family), which moves to the mitochondria, causing the release of cytochromeC. Classical, caspase-dependent apoptosis occurs through the intrinsic pathway [9,10,11], which is initiated when cellular stresses, including heat shock, DNA damage and oxidative stress causes mitochondria to release cytochrome c or Smac/DIABLO to the cytosol. The release of cytochrome c may be attained by members of the Bcl2 protein family, as in the extrinsic pathway. Cytochrome c then leads to the formation of a high molecular weight, apoptosome, which is a caspase-activating complex and is a heptamer comprised of seven Apaf-1 adaptor molecules, each bound to one molecule of cytochrome c, and a dimer of the initiator caspase-9. This, in turn, results in the activation of caspase-9 followed by cleavage and activation of caspases-3 and -7, the effector caspases. Both of these pathways are characterized by less compact chromatin condensation and fragmentation, DNA cleavage with ladder formation and phosphatidylserine exposure associated with membrane blebbing. DNA damage and other stress signals may trigger an increase of p53 proteins, which have three major functions: growth arrest, DNA repair and apoptosis (cell death). The growth arrest stops the progression of the cell cycle, preventing replication of damaged DNA.; (2) Other forms of programmed cell death *viz*. Autophagy (an evolutionary ancient and highly conserved catabolic process in which cytoplasmic material is cloistered in double-membrane vesicles and self-degraded after delivering to the lysosome) [24,25], Anoikis (apoptosis is caused when contact with the extracellular matrix is lost) [26,27], Pyroptosis (commonly triggered by infection with intracellular pathogens causing lytic-programmed cell death and is highly inflammatory) [28,29], Necroptosis (programmed necrosis that occurs without the activation and presence of both caspase and chromatin condensation and is caused when plasma membrane integrity is lost and is followed by phosphorylation of the pseudokinase-mixed lineage kinase domain like MLKL/pMLKL mediated by receptor-interacting kinase 3 RIPK3) [30,31,32], Ferroptosis (iron dependent cell death which is characterized by accumulation of lipid peroxides and subsequent membrane damage) [33,34,35]; (3) Necrosis, which is a passive, uncontrolled cell death process that is characterized by cellular swelling and broken membranes, with random and smeared DNA fragmentation and cell lyses eliciting an inflammatory reaction [36] and (4) Caspase-independent programmed cell death (which is associated with partial chromatin condensation, no DNA laddering, nuclear shrinkage but no fragmentation, abundant autophagosomes, vacuolated cytoplasm and ragged plasma membrane) [17,37].

## 2. Molecular Pathways of Caspase-Independent Programmed Death of Cell

Caspase-independent cell death CI-PCD had been originally considered in the fission yeast *Schizosaccharomyces pombe,* where the yeast genome was without any caspase genes. This observation proved that during evolution, advent of cell death by caspase-independent pathways might have happened prior to caspase-dependent classical apoptosis pathways [38]. However, recently, it has been highlighted that cyanobacterial PCD helped in understanding the incipient mechanism of PCD morphotype(s), about the origin of eukaryotic PCD [39]. Moreover, extensive research on metacaspases in relation to its versatile functions have been reported in plants and non-metazoans, which are also significant enough to highlight its origin [40]. Intracellular molecular factors like TNF-α, Fas/APO-1, IFN-γ, molecular Iodine (I_2_), nitric oxide and CD47 have been reported to activate programmed cell death by caspase-independent pathway [41,42,43,44,45,46,47]. Another cellular extract, IMMUNEPOTENT C-reactive protein (I-CRP) that acts as an immunomodulator, is a Bovine dialysable leukocyte extract which induced cell cycle arrest in the G2/M phase, causes mitochondrial damage and ROS-mediated caspase-independent cell death in HeLa cells [48]. As with the caspase-dependent cell death pathway, the key regulator of programmed cell death in the absence of caspase is the mitochondrion, which, in turn, causes a sequence of events. Apoptotic stimulus causes translocation of the pro-apoptotic protein, Bax, from cytoplasm to mitochondrion. This translocation is caused by the activation of proteases, calpains and cathepsins. It is the lysosomes that release cathepsins and calpains, which are triggered, following the influx of Ca^2+^ in the cell activated by stressed endoplasmic reticular ER [48]. Furthermore, cleavage and translocation of Bid from cytoplasm to the mitochondrion and the cleavage of AIF (apoptotic inducing factor) also required the involvement of cathepsins and calpains. It is the Bax and cleaved Bid at the mitochondrial membrane that cause loss of mitochondrial membrane potential, which, in turn, causes the release of cleaved AIF (tAIF) and Smac/DIABLO due to increased membrane permeability. Cleaved AIF ultimately translocates to the nucleus, and together with endonuclease G, causes condensation of chromatin and fragmentation of DNA. Smac/DIABLO, on the other hand, neutralizes apoptotic inhibitory proteins [46,47]. The molecular pathway has been schematically represented in Figure 2. Caspase-independent cell death occurs under different circumstances. The drug efflux protein, P-glycoprotein, mediated MDR (Multidrug resistance), often expresses caspase-independent cell death triggered by Granzyme B and pore-forming proteins or intact NK cells [49,50,51]. Additionally, even excessive expression of Bcl-xL (anti-apoptotic proteins) and Bcl-2 make the tumor cells tremendously resistant to the diverse cytotoxic stresses caused by irradiation and cytotoxic drugs [41,52]. Furthermore, cell death induced by oncogenic Ras in the absence of caspase activation, is not inhibited by the overexpressing Bcl-2 protein [53]. Even overexpression of Bax often induces cell death where caspase is not required [54]. Bcl-2 and Bcl-xL proteins protect cell from both the types of cell death pathways, caspase-dependent and independent [41,52]. Again, the anti-tumor effect of cytotoxic T lymphocytes (CTL) is executed through two different pathways. The first pathway, which is caspase-dependent, is where the target cell death is triggered through CD95-CD95L ligation on the surface cytotoxic T lymphocytes. The second pathway includes exocytosis of preformed granules which act as lytic effectors, capable of processing caspases and initiating apoptosis [55]. The mechanism involves the insertion of perforin, a pore forming protein, into the target cell membrane, which, in turn, facilitates entry and release of the serine proteases, granzymes (Gr), from endosomal compartments. Interestingly, nuclear apoptosis induced by granzyme B, GrB, depends completely on caspase activation, whereas, GrA-induced nuclear condensation and DNA damage are independent of caspase activation [55,56]. The CD47 antigen, which is prominently expressed on hematopoietic and nonhematopoietic cells, has also been reported to induce cell death in normal and leukemic cells. The CD47 antigen is caspase independent and was defined by exposure of phosphatidylserine (PS), cell shrinkage and mitochondrial matrix swelling in complete absence of nuclear degradation [42]. Thus, tumor cells can be killed by some alternative pathways bypassing the requirement for caspases.

## 3. Molecular Agents Inducing Caspase-Independent Programmed Cell Death

### 3.1. Synthetic Molecules

Several synthetic molecules have been reported to induce cancer cell death without involving caspases (*vide infra*). Apoptosis caused by two kinds of tetracycline analogues TCNAs *viz.* COL-3 (chemically modified tetracycline-3; 6-dimethyl, 6-deoxy, 4-dedimethylamino tetracycline) and doxyclycline DOXY, were studied in colon cancer cell line HT29 and evaluated to induce mitochrondria-mediated apoptosis through both caspase-dependent and caspase-independent pathways [57]. Both of the TCNAs have been reported to act in a dose- and time-dependent manner and inhibit the proliferation of six different colorectal cancer cell lines. COL-3 activated the caspase-dependent apoptotic pathway, whereas DOXY predominantly induced caspase-independent pathway. Inhibitory changes were observed by 10 μg/mL COL-3 and 20 μg/mL DOXY. COL-3 had a stronger effect on the cancer cells than DOXY. COL-3 produced the increase in cytosolic cytochrome c and decrease in mitochondrial membrane potential after 3 h of treatment, and thereafter, activated caspases. While in the case of DOXY, these changes were observed after 24 h of treatment. Bax translocation was not a prerequisite for cytochrome c releasing in COL-3 treatment. Pretreated pancaspase inhibitor (Z-VAD-FMK) reduced COL-3 and DOXY mediated apoptosis up to 81.3 and 35.3%, as compared with nontreated cells, respectively. Endonuclease G and apoptosis-inducing factor were also released into cytosol after the treatment of TCNAs, which indicated that the caspase-independent apoptotic pathway is also one of the key mechanisms for the treatment of TCNAs. Taken together, TCNAs could have strong potentials as antiproliferative drugs in treating colorectal cancers. Another synthetic drug, Bobel 24/AM-24 (2,4,6-triiodophenol) and derivatives have been reported to follow caspase-independent pathways in leukemia and several human pancreatic carcinoma cell lines (NP18, NP9, NP31, and NP29). This mechanism may overcome poor prognosis and resistance to apoptosis observed in pancreatic carcinoma [58]. Except for the NP9 cell line, these compounds induced cytotoxicity and DNA synthesis inhibition, inducing apoptosis in all cell lines. Caspase and/or poly(ADP-ribose) polymerase-1 (PARP-1) activity inhibition was observed in NP18 and a caspase-independent process was shown in NP9. While, in the NP29 and NP31 cell lines, both caspase-dependent and caspase-independent cell death mechanisms coexisted. Cell death was associated with reactive oxygen species (ROS) production, mitochondrial depolarization, cytochrome c and apoptosis-inducing factor (AIF) release, AIF nuclear translocation and lysosomal cathepsin release. Inhibition of ROS production, mitochondrial pore permeability, PARP-1, or phospholipase A2 partially prevented cell death. Hepatocellular carcinoma is also highly chemoresistant to currently available chemotherapeutic agents. A synthetic 6,7-substituted 2-phenyl-4-quinolone, CHM-1, (2′-fluoro-6,7-methylenedioxy-2-phenyl-4-quinolone), has been identified as a selective and potent anti-mitotic anti-cancer agent for human HCC (hepatocellular carcinoma) that acts without activation of the caspase cascade [59]. CHM-1 induced growth inhibition in a concentration-dependent manner in cell lines *viz*, HA22T, Hep3B and HepG2. It interacted with tubulin at the colchicine-binding site and inhibited tubulin polymerization thereby causing microtubular disorganization. CHM-1 caused cell cycle arrest at the G2-M phase by activating Cdc2/cyclin B1 complex activity. However, cyclin-dependent kinase inhibitor significantly reduced CHM-1-induced cell death, activation of Cdc2 kinase activity and increase of MPM2 (mitotic protein monoclonal 2) phosphoepitopes. CHM-1 did not control the caspase cascade and is reported to remain unaffected by pan-caspase inhibitor Z-VAD-FMK. However, CHM-1 induced the translocation of apoptosis-inducing factor (AIF) from mitochondria to the nucleus. Small interfering RNA targeting AIF substantially weakened CHM-1-induced AIF translocation. Notably, CHM-1 has been studied to inhibit tumor growth and prolonged the lifespan in an in vivo model inoculated with HA22T cells. Apparently, the synthetic molecule activates the caspase-independent pathway by triggering the AIF translocation from the mitochondria to the nucleus.

### 3.2. Inorganic Molecules

The inorganic compound, Arsenic trioxide (As_2_O_3_), uses both caspase-dependent and caspase-independent cell death pathways in myeloma cells [60]. It causes caspase-independent cell death in cutaneous Tcell lymphoma cell lines: HuT-78, SeAx, Myla, and in peripheral blood mononuclear cells from patients with Sézary syndrome, triggered by ascorbic acid (vitamin C) [61]. The viability of the cutaneous T cell lymphoma was studied by different biochemical assays *viz*. propidium iodide-annexin-V dual staining, terminal deoxyuridine triphosphate nick end labeling (TUNEL), cell cycle analysis, mitochondrial transmembrane potential alterations, cytochrome c release and detection of processed caspase-3. Both in vitro and in vivo effects of As_2_O_3_ induce apoptosis in a time and concentration dependent manner. However, ascorbic acid of 100 μM potentiated As_2_O_3_-induced Sézary cell death, whereas IFN-α interferon-alpha had no collaborative effect. As_2_O_3_-induced Sézary cell death involved activation of caspase-3, cleavage of poly(ADP-ribose)polymerase and cytochrome c release, but was only partially inhibited by the pancaspase inhibitor Z-VAD fluoromethylketone. Hence, these results demonstrate that As_2_O_3_ synergizes with ascorbic acid to induce Sézary cell death at clinically attainable concentrations through a partially caspase-independent pathway and provide a justification for further in vivo studies. However, recently, sodium arsenite and arsenic trioxide have been reported to execute necroptotic cell death of the L929, mouse fibrosarcoma cell line via a RIP3 receptor-interacting protein 3-dependent pathway [62]. Both the pathways are complementary in cadmium-induced apoptosis in the rat proximal tubular cells [63]. Here BNIP-3(Bcl-2/adenovirus E1B 19-kDa interacting protein 3) acts as an upstream factor, inducing translocation of apoptotic-inducing factor and endonuclease G. Besides, cadmium also induces mitochondria-ROS pathway in MRC-5 lung fibroblasts and causes caspase-independent cell death [64]. It causes caspase-independent apoptosis at concentrations ranging from 25 to 150 μM, which was controlled by reactive oxygen species (ROS) scavengers, such as N-acetylcysteine (NAC), mannitol and tiron. Consistently, the intracellular hydrogen peroxide (H_2_O_2_) was 2.9 fold raised after 3 h of cadmium treatment and diminished quickly within 1 h, as detected with 2,7-dichlorodihydrofluorescein diacetate (DCFH-DA) staining. Using oligomycin A and rotenone, inhibitors of the mitochondrial electron transport chain and mitochondrial permeability transition pore (cyclosporin A and aristolochic acid), it was found circumstantially that ROS production, mitochondrial membrane depolarization and apoptotic content were almost completely or partially stopped. Moreover, as was also revealed by confocal microscopy staining and an anti-AIF antibody, the downfall of mitochondrial membrane potential induced by cadmium was an introduction to the translocation of caspase-independent pro-apoptotic factor, AIF, into the nucleus. Molecular iodine also induces caspase-independent apoptosis in breast carcinoma cell lines *viz.*, MCF-7, MDA-MB-231, MDA-MB-453, ZR-75-1 and T-47D, involving the mitochondria-mediated pathway. The apoptosis mechanism includes DNA fragmentation analysis confirmed by inter-nucleosomal DNA degradation. Terminal deoxynucleotidyl transferase-mediated dUTP nick-end labeling established that iodine induced apoptosis in a time- and dose-dependent manner in MCF-7 cells. Iodine dissipated mitochondrial membrane potential, exhibited antioxidant activity, and caused reduction in total cellular thiol content. Western blot results showed a decrease in Bcl-2 and up-regulation of Bax, and a study by Immunofluorescence confirmed the activation and mitochondrial membrane localization of Bax. Ectopic Bcl-2 overexpression did not prevent iodine-induced cell death. Iodine treatment causes the translocation of apoptosis-inducing factor from mitochondria to the nucleus, and treatment of N-acetyl-L-cysteine prior to iodine exposure restored basal thiol content and ROS levels, and completely inhibited nuclear translocation of apoptosis-inducing factor and subsequently cell death, indicating that thiol depletion may play an important role in iodine-induced cell death [46].

### 3.3. Natural Compounds

Furthermore, many natural compounds also have antitumorigenic properties by caspase-independent programmed cell death pathway. For example, Vitamin D and its two analogues, EB 1089 and CB 1093, which activate CIPCD in breast cancer cell lines, MCF7 and T 47D, by activating cathepsin D and are inhibited by Bcl-2, but do not require p53 or the caspases [65]. The signaling pathways mediating the events of apoptosis induced by these natural compounds cause growth arrest followed by apoptosis in both the cell lines at concentrations varying from 1 to 100 nM, where p53 is not involved for growth-inhibitory effects. Unexpectedly, apoptosis induced by these compounds is independent of caspases and inhibition of caspase activation by different caspase inhibitors showed no effect on the induction of growth arrest or apoptosis by these analogue compounds. Moreover, overexpression of caspase-3 in MCF-7 cells had no effect on vitamin D compounds, and neither caspase-3-like protease activity nor cleavage of a caspase substrate poly(ADP)ribose polymerase was detected from apoptotic cells following the treatment with these compounds. Overexpression of an anti-apoptotic protein, Bcl-2, in MCF-7 cells further conferred a protection from apoptosis induced by vitamin D compounds. An alkaloid, staurosporine, isolated from *Streptomyces staurosporeus*, exhibits anti-cancer activity by Cathepsin D mediating cytochrome c released in human fibroblast. Cathepsin D triggers Bax activation, resulting in relocation of apoptosis inducing factor (AIF) in T-lymphocytes, leading to apoptosis [66,67]. Activated human T lymphocytes exposed to apoptotic stimuli by staurosporine showed early caspase-independent phase characterized by cell shrinkage and peripheral chromatin condensation. During this phase, AIF is released from the intermembrane space of mitochondria, Bax undergo conformational changes, relocation to mitochondria, and insertion into the outer mitochondrial membrane in a Bid-independent manner. The subcellular contribution of different cathepsins for caspase-independent factors responsible for Bax activation and AIF release was analyzed. Cathepsins were translocated from lysosomes to the cytosol. Thus, the novel sequence of events involves cathepsin triggers Bax activation, Bax induces the selective release of mitochondrial AIF, and the latter is responsible for the early apoptotic changes. The functional glycoprotein, thrombospondin-1 triggers caspase free cell death in promyelocytic leukemia NB4 cells and freshly isolated monocytes and monocyte-derived dendritic cells through thrombospondin-1 membrane receptors CD47 and α vβ3 [68]. Other triggering events were characterized by the instantaneous permeability of the plasma membrane, exposure of phosphatidylserine, decreased mitochondrial membrane potential and by fragmented DNA sequences [67]. Though mitochondria membrane depolarization was essential to thrombospondin-1action, it did not release death-promoting proteins like non-caspase apoptosis regulators, apoptosis-inducing-factor AIF, endonuclease G, or caspase regulators, cytochrome c or second mitochondrial activator of caspase/direct inhibitor of apoptosis protein-binding protein with low isoelectric point Smac/DIABLO. Reactive oxygen species (ROS) production was also not involved in the death process. Another natural compound, selenite, was capable of inducing CIPCD in the cervical cancer cell lines HeLa and Hep 2 cell lines. The pathway involves oxidative stress-mediated activation of p53 and p38, accumulation of Bax and release of AIF and Smac/DIABLO without activation of caspases. Despite this, even with co-treatment with the caspase inhibitor, Z-VAD-FMK, cell death was observed [69]. Selenite at concentrations of 5 and 50 μmol/L during 24 h of exposure in the HeLa and Hep-2 cell lines produced time- and dose-dependent suppression of DNA synthesis and induced DNA damage, which resulted in phosphorylation of histone H2A.X. Following the DNA damage, selenite activated the p53-dependent pathway, as evidenced by the appearance of phosphorylated p53 and the accumulation of p21 in the treated cells. Additionally, selenite also activates the p38 pathway. Ultimately, the p53- and p38-dependent signaling pathways led to the accumulation of Bax protein, which is preventable by specific inhibitors of p38 and p53. During the process, mitochondria also changed their dynamics and released AIF and Smac/Diablo, which initiated caspase-independent apoptosis. Lipoic acid elicits its anti-tumor effects via CI-PCD in HL-60 leukemia cells [70]. Lipoic acid inhibits both cell growth and viability in a time- and dose-dependent manner. Interruption of the G1/S and G2/M phases of cell cycle progression accompanied by the induction of apoptosis was observed following the treatment. Evidence supporting the induction of apoptosis was based on the appearance of sub-G1 peak and the cleavage of poly(ADP-ribose) polymerase (PARP). Apoptosis was further preceded by a decrease in the expression of anti-apoptotic protein Bcl-2, increased expression of bax, and the release and translocation of apoptosis-inducing factor AIF and cytochrome c from the mitochondria to the nucleus, without altering the subcellular distribution of the caspases. In a similar manner, a naturally occurring selenoamino acid, selenocystine, induces CIPCD in MCF7 breast cancer cells via translocation of AIF and phosphorylation of p53, and is suggested to be a promising natural anticancer compound [71]. Epidemiological, preclinical and clinical studies reported selenium as a potential cancer chemopreventive and chemotherapeutic agent. Although the mechanisms of apoptosis by selenoamino acid remain indefinable, it was shown that the natural compound induced caspase-independent apoptosis in MCF-7 breast carcinoma cells, which was accompanied by poly(ADP-ribose) polymerase (PARP) cleavage, caspase activation, DNA fragmentation, phosphatidylserine exposure and nuclear condensation. It also induced a decrease of mitochondrial membrane potential by regulating the expression and phosphorylation of Bcl-2 family members [71]. This led to the mitochondrial release of cytochrome c and apoptosis-inducing factor (AIF), which then translocated into the nucleus and induced chromatin condensation and DNA fragmentation. MCF-7 cells treated with selenoamino acid were also shown to exhibit an increase of p53, as well as the silencing and attenuating of p53, which partially suppressed the cell apoptosis. Furthermore, two more upstream cellular events: generation of reactive oxygen species ROS and subsequent induction of DNA strand breaks, were also found. The thiol-reducing antioxidants, N-acetylcysteine and glutathione, were reported to completely block cell apoptosis. Berberine, a natural isoqinoline alkaloid, inhibits cell growth in human pancreatic cancer cells, BxPC-3, mediating through a caspase-independent cell death pathway [72]. It is also reported to stimulate caspase-independent cell death in mouse immorto-Min colonic epithelial cells (IMCE) through production of ROS, leading to the release of cathepsin B and the activation of PARP-dependent AIF translocation [73]. Berberine decreased colon tumor colony formation and induced cell death and lactate dehydrogenase LDH release in IMCE cells. Additionally, normal colon epithelial cells, young adult mouse colonic epithelium YAMC, were not sensitive to berberine. Berberine-induced cell death did not stimulate caspase activation and PARP cleavage and neither were affected by caspase inhibitor in IMCE cells. Rather, it stimulated the release of apoptosis-inducing factor AIF from mitochondria to nucleus in a ROS dependent manner. Berberine-stimulated ROS production leads to cathepsin B release and PARP activation-dependent AIF activation, resulting in caspase-independent cell death in colon tumor cells. Many natural flavonoids, such as quercetine, myricetine and apigenin, are also designated to be specific caspase inhibitors of casp 1, 3 and 7, and induced cell death in MDA-MB-231, an epithelial human breast cancer cell, through a non-classical apoptosis pathway that is not dependent on caspase activity. Hence, they may be a lead source for the rational drug design of caspase-specific inhibitors [74]. However, the mechanism of the protective effect of these natural flavonoids against cancer is not fully understood, it may be related to flavonoids’ ability to inhibit the NF-κB-signaling pathway, where NF-κB activity is thought to suppress apoptosis and promote cancer cell growth and metastasis. Matrine, another natural alkaloid isolated from the root of *Sophora subprostrata*, induces parallelly caspase-dependent and caspase-free cell death in HepG2 cells through a Bid-regulated AIF nuclear translocation pathway [75]. It was further studied by the authors that AIF nuclear translocation, and the subsequent cell death, was prevented by Bid inhibitor BI-6C9, Bid-targeted siRNA and ROS scavenger Tiron. However, the mechanisms involved are still not completely known [75]. The tomato glycoalkaloid, α-Tomatine, induces both in vivo and in vitro caspase-free cell death in CT-26 Cells, an N-nitroso-N methylurethane induced undifferentiated colon carcinoma cell line from BALB/C mouse. This was supported by western blot expression of apoptosis-inducing protein (AIF) localization from mitochondria to nucleus and down-regulation of survivin, an inhibitor of apoptosis. It also failed to express the active form of caspase-3, -8, and -9 produced by proteolytic cleavage in CT-26 cancer cells [76]. The natural compound, resveratrol seems to trigger caspase-independent cell death in MCF-7 breast cancer cells through changes in mitochondrial membrane potential, downregulating Bcl-2, increased ROS (reactive oxygen species) and nitric oxide production and prevention of NF-kB [77]. Interestingly, a derivative of resveratrol, oxyresveratrol (a hydroxyl-substituted stilbene), was also reported to induce apoptosis-like cell death, resulting in an accumulation of cells at the sub-G1 phase of the cell cycle in triple negative breast cancer cells, MDA-MB-231, by caspase-independent pathway through chromatin condensation, induction of ROS, DNA fragmentation, phosphatidyl serine externalization, PARP cleavage, decrease in mitochondrial membrane potential Δψm and nuclear translocation of AIF [78]. Molecular docking studies showed the binding of S1 site of caspase-3 to oxyresveratrol and could also exert an inhibitory effect on this executioner caspase. The morphology and cell viability studies with the pan-caspase inhibitor, QVD-OPH, further revealed caspase-independent cell death induced by oxyresveratrol. The immunoblot analysis demonstrating the absence of caspase cleavage during cell death also confirmed these findings. A natural phenolic compound, curcumin, has been recently reported to induce apoptosis of immortalized human keratinocytes (HaCaT) cells [79]. A 24-μM dose of circumin arrested the cells at the G2/M phase, with an apoptosis rate of 33.95%. HaCaT cells showed changes in typical apoptotic morphology and the configuration of nuclear matrix-intermediate filaments after treatment. 16 differentially expressed nuclear matrix proteins were identified, including apoptosis-inducing factor (AIF) and caspase 3, by 2-DE and MALDI-TOF/TOF mass spectrometry. Immunofluorescence assays further showed that AIF was released from the mitochondria to the nucleus and the expression of AIF increased in the nucleus. However, AIF silencing and caspase inhibitor (Z-VAD-FMK) both lead to HaCaT cells being insensitive to apoptosis induced by curcumin. Meanwhile, after curcumin treatment, mitochondrial membrane depolarization led to cytochrome c release from the mitochondria to the cytoplasm, and the ratio of Bax to Bcl-2 in HaCaT cells was also increased, which subsequently initiated the activation of caspase-3. Taken together, these results suggested that curcumin-induced apoptosis in HaCaT cells occurs through both caspase-independent and caspase-dependent pathways [79]. Apart from the above listed natural compounds, few more natural extracts were also reported to follow both caspase-dependent and caspase-free cell death pathways. A bacterial virulent factor, *Clostridium diffificile* toxin B (TcdB) activates the caspase-dependent and caspase-free apoptosis in HeLa and MCF-7 cancer cell line, respectively [80]. The thiosulfinates from *Allium tuberosum* L. activate both CD-PCD and CI-PCD in prostate cancer cell line PC3, by increasing the expression of Bax and AIF and decreasing the expression of Bcl-2 [81]. Recently, *Chelidonine* *majus* L., family Papaveraceae, containing isoquinoline alkaloids such as sanquinarine, chelidonine, chelerythrine, berberine, protopine and coptisine, flavonoids and phenolic acids as major compounds, were reported to induce both caspase-dependent and caspase-independent cell death in the T98G Human Glioblastoma Cell Line. Like other anti-cancer compounds, Chelidonine also induced apoptosis in a dose-dependent manner. Caspase-3 and -9 were activated by the treatment, but chelidonine-mediated apoptosis was only partially inhibited by pan-caspase inhibitor. Chelidonine also induced the translocation of AIF into the nucleus, therefore, it is likely that chelidonine induces T98G cell death through both caspase-dependent and caspase-independent apoptosis pathways. Chelidonine also induced G2/M arrest and led to cell death by inhibiting mitosis, inducing multipolar spindle assembly. Active Cyclin dependent kinase, CDK1, one of the factors contributing to the prolongation of the G2/M phase, induced Mcl-1 (apoptotic inhibitor) degradation, thereby increasing mitochondrial instability, causing apoptosis in chelidonine-treated T98G cells [82]. Unlike other natural extracts, the rhizome of ginger (*Zingiber officinale*), a common culinary agent, is known for its caspase-independent paraptosis in triple negative breast cancer (MDA-MB-231) and non-small lung (A549) cancer cells via ER stress, mitochondrial dysfunction, AIF translocation and DNA damage [83].

### 3.4. Repurposing Drugs

Several repurposing cancer chemotherapeutic drugs induce caspase-independent cell death in cancer cells. However, the process depends on the inducing agents and the micro-environment of the cancer cells. For example, caspase-inhibitor Z-VAD-FMK exhibited that, in human endothelial cell HUVECs and ovarian cancer cell line A2780, Doxorubicin-induced apoptosis through a caspase-dependent mechanism, where as in neonatal rat cardiac myocytes NeRCaMs, the same molecule has been reported to show caspase-independent mechanisms. It was further confirmed that flavonoid mono hydroxy ethylrutoside, monoHER, a protective agent used against toxicity of Doxorubicin, at different concentrations, acts by suppressing the caspase-dependent or caspase-independent cell death activation of molecular mechanisms [84]. A synthetic N-methylpiperidinyl chlorophenyl flavone, Flavopiridol, prevents both in vitro and in vivo growth of renal, prostate and colon cancers, such as solid malignancies, by decreasing the expression of cyclin D1, CDK4 and p21. This agent induces CI-PCD by down-regulating the release of cytochrome c, Bcl-2 and the translocation of AIF to the nucleus [85,86]. Several other accepted cancer chemotherapeutic drugs, such as cladribine, camptothecin and cisplatin, are also capable of causing cell death in various cancer cells through caspase-independent programmed cell death pathway involving translocation of apoptotic inducing factor AIF from mitochrondrion to nucleus [87,88,89,90,91,92]. Another novel amphiphilic cationic compound, Atiprimod, was reported to display a strong anti-mantle cell lymphoma activity by inducing cell apoptosis mainly via activation of the apoptotic inducing factor pathway [93]. Earlier, paclitaxel poliglumex, which is a phase II clinical trial drug, was reported to be metabolized via cathepsin B to paclitaxel in cancer cells, such as prostate and non-small cell lung carcinoma [94,95]. Recently it has been reported that clinically applicable concentrations of eribulin and the classical agent targeting microtubule, paclitaxel, predominantly cause cell death without activation of caspase in MCF-7 breast carcinoma cells [96]. On the molecular level, both the compounds, to a similar extent, activate the key proteins involved in apoptosis, such as p53, Plk1, caspase-2 and Bim, as well as Mitogen-activated protein kinase pathway (MAPK) mediated by extracellular signal-regulated kinases (ERK) and c-Jun amino-terminal kinases (JNK). Metformin, another repurposing anti-cancer drug, has been recently reported to induce apoptosis in the human bladder cancer cell line, T24, through the stimulation of the AIF signaling pathway and by increasing c-FLIPL protein (FADD like interleukin-1β-converting enzyme inhibitory protein) instability [97]. However, the induced apoptosis was partially inhibited by a general caspase inhibitor, Z-VAD-FMK, which suggested that metformin-induced apoptosis in T24 cells is partially caspase-independent. Furthermore, treatment with the reactive oxygen species scavenger N-acetylcysteine failed to suppress metformin-induced apoptosis and c-FLIPL protein degradation. Though the mechanism is largely unknown, in conclusion, these results suggested that metformin-induced apoptosis was mediated through AIF-promoted caspase-independent, as well as caspase-dependent, pathways in T24 cells [97]. Figure 3 and Table 1 give an overview of caspase-independent cell death induced by different classified chemotherapeutic agents.

## 4. Summary

Recent knowledge based on research suggests that, besides having the caspase-dependent apoptotic pathways, cells have other alternative methods of programmed cell death which are equally promising. Several caspase-mediated drugs have been shown to have low anti-tumorigenic potencies. The low efficiency of such drugs could be accredited to the fact that human tumors often develop protein mutations, involved in the classical caspase-mediated programmed cell death pathways. Furthermore, tumor cells are often resistant to classical caspase-mediated pathways because of excessive expression of anti-apoptotic proteins. Due to these limitations, it is becoming essential to identify and design novel, effective anti-tumor agents based on above-mentioned inorganic, synthetic, natural and repurposing molecules targeting caspase-independent programmed cell death. However, few small molecules have been reported to target both caspase-dependent and caspase-independent pathways in the same in vitro model. Caspase-independent cell death pathways potentially and effectively work in conjunction with caspase-dependent cell death in the above-mentioned cancer cell lines. Molecules targeting these dual pathways will give a better idea for future drug designs. However, further details regarding the molecular mechanism of this alternative pathway are essential, and should be based on in vivo study on animal models. This would give more correct and vital information on the ability of the drug for induction of caspase-independent programmed cell death.

## Figures and Tables

**Figure 1 molecules-27-06401-f001:**
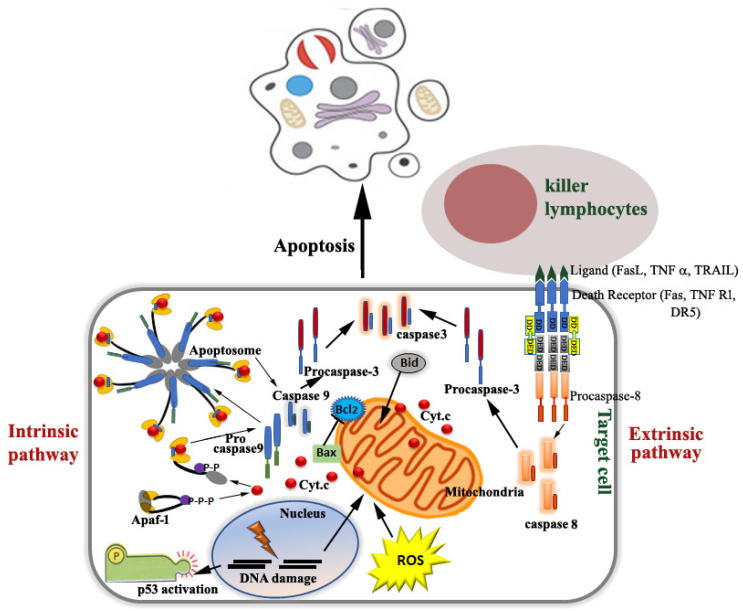
A schematic representation depicting the molecular pathways with the biomarkers of caspase-dependent programmed cell death: intrinsic and extrinsic pathways. [The intrinsic pathway (LHS in the figure) begins inside a cell that is mitochondrial-mediated, where apoptosis is initiated when cellular stresses caused by damage of DNA, shock due to heat and oxidative stress, causing the release of cytochrome c or Smac/DIABLO from the mitochondrial intermembrane space to the cytosol, takes place. Release of cytochrome c is achieved by Bcl2 protein family (Bax, Bid), which translocate to the mitochondria, and/or oligomerize within mitochondrial membranes, forming pores which in turn releases cytochrome c to the cytoplasm from the mitochondrial intermembrane space. Released cytochrome c, in turn, binds to Apaf-1, a cytosolic protein that normally exists as an inactive monomer. The binding of cytochrome c induces a conformational change in Apaf-1, allowing it to bind the nucleotide dATP or ATP. The nucleotide binding to the Apaf-1–cytochrome c complex triggers its oligomerization to form the apoptosome (a high molecular weight heptameric compound). Apoptosome activates caspase-9, which then cleaves, and thereby, activates the effector caspases, caspases-3 and -7. DNA damage and other stress signals may trigger the increase of p53 proteins. The extrinsic pathway (RHS in the figure) begins outside a cell that is receptor-mediated and initiated by receptor ligation, trimerization and recruitment of adapter molecules like FADD and TRADD to the death domain (DD), forming a complex known as the death-inducing signaling complex (DISC). Procaspase-8 then binds to the death effector domains (DED) of the adaptor molecules, which is then followed by oligomerization and activation of caspase-8. Caspase-8 then either directly activates the executioner caspases-3 and -7 or activates the cytosolic protein, Bid, to translocate into the mitochondria, causing the release of cytochrome c].

**Figure 2 molecules-27-06401-f002:**
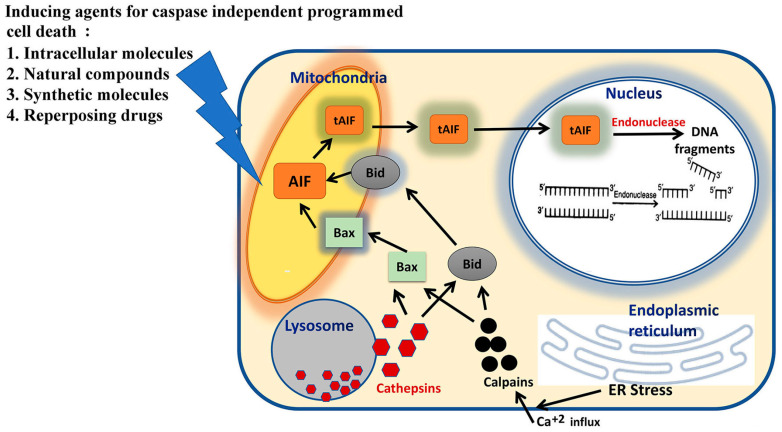
A schematic representation depicting the molecular pathways with the biomarkers of caspase-independent programmed cell death pathway. [Mitochondria is the main regulator which initiates number of events. In response to a stimulus, Bax, the pro-apoptotic protein, translocates to the mitochondrion. This translocation is controlled by cathepsins and calpains activity. Cathepsins are released from the lysosomes and calpains are activated by endoplasmic reticulum ER stress induced Ca^2+^ influx in the cell. Cathepsins and calpains also cause cleavage and translocation of Bid to the mitochondrion and cleavage of AIF. The presence of Bax and cleaved Bid at the mitochondrial membrane induces mitochondrial depolarization, thereby causing increased membrane permeability, which, in turn, causes the release of cleaved AIF (sAIF) and Smac/DIABLO. sAIF translocates to the nucleus where it, jointly with endonuclease G, induces chromatin condensation and DNA fragmentation of high molecular weight fragments of ~50 kb].

**Figure 3 molecules-27-06401-f003:**
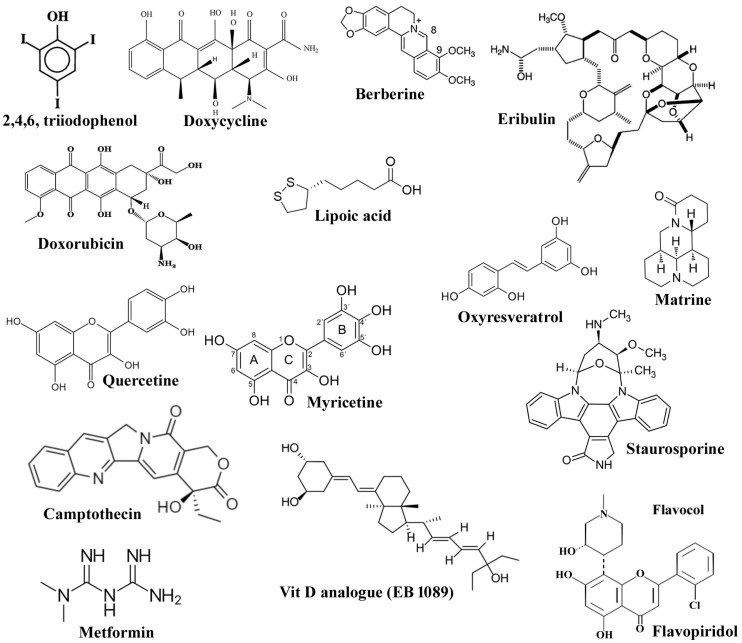
Chemical structures of molecules inducing caspase-independent cell death.

**Table 1 molecules-27-06401-t001:** Overview of inducing agents engaged in caspase-independent programmed cell death.

Classification	Name of the Compounds	System; Triggering Pathway	References
**Intracellular molecular factors/extract**	TNF-α, CD47, IMMUNEPOTENT C-reactive protein (I-CRP)	Human neutrophils, leukemic cells, Hela cells; induced caspase free cell death pathway.	[42,44,48]
**Synthetic molecules**	Tetracycline analogues: TCNAs, COL-3 (chemically modified tetracycline-3; 6-demethyl, 6-deoxy, 4-dedimethylamino tetracycline) and doxyclycline DOXY	Colon cancer cell line HT29; evaluated to be mitochondria-mediated apoptosis through both caspase-dependent and -independent pathways.	[57]
	Bobel-24/AM-24 (2,4,6-triiodophenol) and derivatives	Human pancreatic carcinoma cell lines (NP18, NP9, NP31, and NP29), leukemia cell; follows caspase-independent pathways via production of ROS, mitochondrial depolarization, release of cytochrome c, AIF and release of lysosomal cathepsin.	[58]
	6,7-substituted 2-phenyl-4-quinolone, CHM-1, (2′-fluoro-6,7-methylenedioxy-2-phenyl-4-quinolone)	HCC (hepatocellular carcinoma) HA22T, Hep3B, and HepG2; selective and potent anti-cancer agent and acts without activation of the caspase cascade.	[59]
**Inorganic compounds**	Arsenic trioxide As_2_O_3_	Myeloma cells and cutaneous T cell lymphoma *viz.* HuT-78, SeAx, Myla, and of peripheral blood mononuclear cells; follows both caspase-dependent and caspase-independent cell death pathways, caspase-independent cell death is triggered by ascorbic acid (vitamin C).	[60,61,62]
	Cadmium	Rat proximal tubular cell, MRC-5 fibroblasts; Both caspase-dependent and caspase-independent pathways are caused that acts synergistically. BNIP-3 (Bcl-2/adenovirus E1B 19-kDa interacting protein 3) acts as an upstream factor inducing translocation of AIF and endonuclease G. It also induces mitochondria-ROS pathway and causes caspase-independent cell death.	[63,64]
	Molecular iodine	Breast carcinoma cell lines *viz*., MCF-7, MDA-MB-231, MDA-MB-453, ZR-75–1, and T-47D; caspase-independent apoptosis involving the mitochondria-mediated pathway.	[46]
**Natural compounds**	Vit D	Breast cancer cell lines: MCF7, T47D; activates CIPCD by activating cathepsin D and inhibited by Bcl-2 but does not require p53.	[65]
	Staurosporine	Human fibroblast; exhibits anti-cancer activity by Cathepsin D mediating cytochrome c release. Cathepsin D triggers Bax activation, resulting in relocation of AIF in T-lymphocytes, leading to apoptosis.	[66,67]
	Thrombospondin-1	Promyelocytic leukemia NB4 cells and freshly isolated monocytes and monocyte-derived dendritic cells through Thrombospondin-1 membrane receptors CD47 and αvβ3, triggered caspase free cell death and characterized by the instantaneous permeability of plasma membrane, exposure of phosphatidylserine, decreased mitochondrial membrane potential and highly fragmented DNA.	[68]
	Selenite	Cervical cancer cell lines HeLa and Hep 2 cell lines; The Caspase free programmed cell death pathway involves activation of p53, accumulation of Bax and release of AIF and Smac/DIABLO, co-treatment with the caspase inhibitor Z-VAD-FMK, cell death was observed.	[69]
	Lipoic acid	HL-60 leukemia cells; activates CI-PCD via up-regulation of Bax, downregulating Bcl-2, release and translocation of AIF and cytochrome c to nucleus from mitochondria.	[70]
	Selenocystine	Breast cancer cells MCF7; CIPCD via translocation of AIF and phosphorylation of p53.	[71]
	Berberine	Human pancreatic cancer cells, BxPC-3, mouse immorto-Min colonic epithelial cells (IMCE), normal colon epithelial cells, namely young adult mouse colonic epithelium (YAMC) cells; stimulate caspase-independent cell death through production of ROS leading to the release of cathepsin B and activation of PARP dependent AIF translocation.	[72,73]
	Natural flavonoids: Quercetine, Myricetine, Apigenin	MDA-MB-231, an epithelial human breast cancer cells; induced cell death through a non-classical apoptosis pathway that is not dependent on caspase activity. Hence, they may be lead source for the rational drug design of caspase specific inhibitors.	[74]
	Matrine	HepG2 cells; induces parallelly caspase-dependent and caspase free cell death through Bid regulated AIF nuclear translocation pathway.	[75]
	α-Tomatine	Mouse colon cancer cells CT-26; induces both in vitro and in vivo caspase free cell death by expression of apoptosis-inducing protein (AIF) localizes from mitochondria to nucleus and down-regulation of surviving, an inhibitor of apoptosis. It also failed to express the active form of caspase-3, -8, and -9 produced by proteolytic cleavage.	[76]
	Resveratrol, derivative- Oxyresveratrol	MCF-7 breast cancer cells, MDA-MB-231 breast cancer cells; trigger the caspase-independent cell death through changes in mitochondrial membrane potential, downregulating Bcl-2, increased ROS and nitric oxide production and prevention of NF-kB. The derivative compound, induces apoptosis-like cell death by caspase-independent pathway through the induction of ROS, DNA fragmentation, Phosphatidyl serine externalization, PARP cleavage, decrease in mitochondrial membrane potential Δψm and nuclear translocation of AIF.	[77,78]
	Curcumin	Human keratinocytes (HaCaT); induced apoptosis of the cells through both caspase-dependent and caspase-independent pathways.	[79]
**Natural extracts**	*Clostridium diffificile* toxin B (TcdB)	HeLa and MCF-7 cancer cell line; activates caspase-dependent and caspase-free apoptosis, respectively.	[80]
	The thiosulfinates from *Allium tuberosum* L. extract	Prostate cancer cell line PC3; activate both CD-PCD and CI-PCD by increasing the expression of Bax and AIF and decreasing the expression of Bcl-2.	[81]
	Extract of *Chelidonine majus* L., containing isoquinoline alkaloids like sanquinarine, chelidonine, chelerythrine, berberine, protopine and coptisine, flavonoids and phenolic acids	T98G Human Glioblastoma Cell Line; reported to induce both caspase-dependent and caspase-independent cell death through G2/M arrest.	[82]
	The rhizome of ginger (*Zingiber officinale*)	Triple negative breast cancer (MDA-MB-231) and non-small lung (A549) cancer cells; known for its caspase-independent paraptosis via ER stress, mitochondrial dysfunction, AIF translocation and DNA damage.	[83]
**Repurposing drug molecules**	Flavopiridol	Glioma cell lines; independent of retinoblastoma and p53 tumor suppressor pathway alterations by a caspase-independent pathway.	[85,86]
	Doxorubicin	Human endothelial cell HUVECs and ovarian cancer cell line A2780, neonatal rat cardiac myocytes NeRCaMs; induced apoptosis through caspase-dependent and caspase-independent mechanisms, respectively. It was further confirmed that flavonoid mono hydroxy ethylrutoside, monoHER, a protective agent used against toxicity of Doxorubicin, at different concentrations, acts by suppressing the caspase-dependent or -independent cell death activation.	[84]
	Cladribine, camptothecin and cisplatin	Hepatocellular cancer cell line, Human SKOV3 ovarian carcinoma cells, LNCaP prostate cancer cells; capable of causing cell death through caspase-independent programmed cell death pathway involving translocation of AIF from mitochondrion to nucleus.	[87,88,89,90,91,92]
	Atiprimod	Mantle cell lymphoma MCL; inducing cell apoptosis mainly via activation of AIF pathway.	[93]
	Paclitaxel poliglumex	Prostate, ovarian cancer and non-small cell lung carcinoma; is a phase II clinical trial drug, metabolized via cathepsin B to paclitaxel in the cancer cells. In addition, it also induced caspase-independent apoptosis via apoptosis inducing factor AIF.	[94,95]
	Eribulin, Paclitaxel	MCF-7 breast carcinoma; predominantly causes cell death without activation of caspase. On the molecular level, both the compounds to a similar extent activate the key proteins involved in apoptosis such as p53, Plk1, caspase-2, and Bim as well as MAPK pathway mediated by ERK and JNK.	[96]
	Metformin	Human bladder cancer cell line T24; induces apoptosis by both caspase-dependent and caspase-independent signaling pathways through the stimulation of AIF signaling pathway and increasing c-FLIPL protein (FADD like interleukin-1β-converting enzyme inhibitory protein) instability.	[97]

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
