# Peer review of "A Mini Review on Molecules Inducing Caspase-Independent Cell Death: A New Route to Cancer Therapy"

_molecules, 2022, doi:10.3390/molecules27196401_

Round 1

Reviewer 1 Report

Bibliographic research should be improved.

Author Response

Dear Sir/ Madam,

I am resubmitting the MS and are extremely thankful to the reviewer for his/her interest in my work and for critical and judicious evaluation of the manuscript. As indicated below, I have checked the comments and have made necessary changes (highlighted with yellow colour) according to the suggestion.

Title: “A mini review on molecules inducing caspase independent cell death: a new role to cancer therapy”

Manuscript No. Molecules: 1903472

Answers to the points raised by the reviewer are as follows:

Report 1: all corrections have been incorporated and highlighted with yellow colour.

Reviewer 2 Report

The manuscript entitled ‘A mini review about molecules inducing caspase independent cell death: a new role to cancer therapy’ evaluates several agents inducing cancer cell death, analysing their different molecular pathways. The main focus is the caspase independent programmed cell death, of which several examples are reported. Despite the large amount of informations collected, several modifications need to be made to improve the quality of the article. Hence, the manuscript is suitable for publication after major revisions.

General comments to this review are:

-      The manuscript needs to be grammatically checked by an expert. English should be revised and frequent typing error are also present.

-    The abstract doesn’t need to be divided in sections (background, aims, results ..).

-     In the introduction, the double classification makes the text redundant. To avoid repetitions, authors should just describe the second one, since most complete.

-      Please rephrase lines 135-137.

-  Paragraph 3 needs to be reorganized, dividing the molecules in subparagraphs (intracellular molecules, natural compounds, synthetic molecules, repurposing drugs) as reported in figure 2.

-   At the end of line 185, the please add some appropriate citations. In particular: European Journal of Medicinal Chemistry 237:114399 DOI: 10.1016/j.ejmech.2022.114399; Eur J Med Chem. 2022 Mar 16; 235:114292. doi: 10.1016/j.ejmech.2022.114292; ACS Med. Chem. Lett. 2022, 13, 3, 358 - 364 https://doi.org/10.1021/acsmedchemlett.1c00600. The appropriate citation is also missing in line 135 and 365.

-   Please add a column in table 1, inserting the chemical structures for each row.

Author Response

Dear Sir/ Madam,

I am resubmitting the MS and are extremely thankful to the reviewer for his/her interest in my work and for critical and judicious evaluation of the manuscript. As indicated below, I have checked the comments and have made necessary changes according to the suggestion.

Title: “A mini review on molecules inducing caspase independent cell death: a new role to cancer therapy”

Manuscript No. Molecules: 1903472

Answers to the points raised by the reviewer are as follows:

Report 2:The manuscript entitled ‘A mini review about molecules inducing caspase independent cell death: a new role to cancer therapy’ evaluates several agents inducing cancer cell death, analysing their different molecular pathways. The main focus is the caspase independent programmed cell death, of which several examples are reported. Despite the large amount of informations collected, several modifications need to be made to improve the quality of the article. Hence, the manuscript is suitable for publication after major revisions.

General comments to this review are:

-      The manuscript needs to be grammatically checked by an expert. English should be revised and frequent typing error are also present.

Corrections incorporated.

-    The abstract doesn’t need to be divided in sections (background, aims, results ..).

Done as per suggestion

-     In the introduction, the double classification makes the text redundant. To avoid repetitions, authors should just describe the second one, since most complete.

corrected

-      Please rephrase lines 135-137.

corrected

-  Paragraph 3 needs to be reorganized, dividing the molecules in subparagraphs (intracellular molecules, natural compounds, synthetic molecules, repurposing drugs) as reported in figure 2.

Divided into subparagraphs

-   At the end of line 185, the please add some appropriate citations. In particular: European Journal of Medicinal Chemistry 237:114399 DOI: 10.1016/j.ejmech.2022.114399; Eur J Med Chem. 2022 Mar 16; 235:114292. doi: 10.1016/j.ejmech.2022.114292; ACS Med. Chem. Lett. 2022, 13, 3, 358 - 364 https://doi.org/10.1021/acsmedchemlett.1c00600.

I have downloaded all the pdfs. All are excellent synthetic molecules with high anti-cancer efficacies. But I am not sure (could not find in the pdfs) whether the molecules follow caspase independent pathways. The sentence read as….. “Several synthetic molecules have been reported to induce cancer cell death without involving caspases”.

Hence the references were not included.

The appropriate citation is also missing in line 135 and 365.

Incorporated.

-   Please add a column in table 1, inserting the chemical structures for each row.

Because of lack of space in the column of the Table, I have included a separate figure as Fig 3 that showed few anti cancer molecules following CI-PCD.

Round 2

Reviewer 2 Report

Appropriate citations (suggested in the previous revision) have not been added. Furthermore, the chemical structures of some compounds reported in table 1 are still missing.